# Embracing Growth, Adaptability, Challenges, and Lifelong Learning: A Qualitative Study Examining the Lived Experience of Early Career Nurses

**DOI:** 10.3390/nursrep15060214

**Published:** 2025-06-12

**Authors:** Liz Ryan, Di Stratton-Maher, Jessica Elliott, Tracey Tulleners, Geraldine Roderick, Thenuja Jayasinghe, Joanne Buckley, Jamie-May Newman, Helen Nutter, Jo Southern, Lisa Beccaria, Georgina Sheridan, Danielle Gleeson, Haiying Wang, Sita Sharma, Jing-Yu (Benjamin) Tan, Linda Ng, Blake Peck, Tao Wang, Daniel Terry

**Affiliations:** 1School of Nursing and Midwifery, University of Southern Queensland, Toowoomba, QLD 4350, Australia; liz.ryan@unisq.edu.au (L.R.); dianne.stratton-maher@unisq.edu.au (D.S.-M.); jessica.elliott@unisq.edu.au (J.E.); tracey.tulleners@unisq.edu.au (T.T.); geraldine.roderick@unisq.edu.au (G.R.); thenuja.jayasinghe@unisq.edu.au (T.J.); joanne.buckley@unisq.edu.au (J.B.); jamie-may.newman@unisq.edu.au (J.-M.N.); helen.nutter@unisq.edu.au (H.N.); jo.southern@unisq.edu.au (J.S.); lisa.beccaria@unisq.edu.au (L.B.); georgina.sheridan@unisq.edu.au (G.S.); danielle.gleeson@unisq.edu.au (D.G.); emily.wang@unisq.edu.au (H.W.); sita.sharma@unisq.edu.au (S.S.); benjamin.tan@unisq.edu.au (J.-Y.T.); b.peck@federation.edu.au (B.P.); daniel.terry@unisq.edu.au (D.T.); 2Centre for Health Research, University of Southern Queensland, Springfield, QLD 4300, Australia; 3Institute of Health and Wellbeing, Federation University Australia, Ballarat, VIC 3350, Australia

**Keywords:** nurses’ role, professional practice, education, nursing, continuing, staff development, policy making

## Abstract

**Background:** Healthcare is a dynamic environment for nurses, with early career nurses (ECNs) needing to adapt and learn while also meeting care demands. Effective support systems, mentorship, and continuous professional development are vital in facilitating their transition while navigating competing demands. The aim of this study is to interpret and understand the meaning of ECNs’ professional experiences four years after completing their bachelor’s degree in Australia. **Method:** A qualitative descriptive design using a hermeneutic phenomenological approach was used as part of a longitudinal study. Follow-up semi-structured interviews were conducted among twenty-five ECNs between 2022 and 2024 using purposive sampling to recruit ECNs who had graduated four years ago. Thematic analysis was used to analyse data while adhering to the consolidated criteria for reporting qualitative research (COREQ) guidelines. **Results:** Four themes emerged among participants, which encompassed professional growth and unwavering commitment, ongoing professional adaptability, feeling stuck with limited choices, and continual learning amid career challenges and personal life demands. **Conclusions:** Change is needed to ensure professional learning becomes a shared responsibility among policy makers and healthcare leaders and to ensure that professional learning leads to more nurses taking up further study, thus increasing the safety and quality of care delivered in the healthcare environment.

## 1. Introduction

The nursing profession is inherently dynamic, requiring continuous adaptation and learning to meet evolving healthcare demands. Career pathways in nursing are marked by significant professional growth, adaptability, and challenges, particularly for early career nurses (ECNs) transitioning from tertiary education to professional practice. This transition for ECNs is often marked by a steep learning curve and the need for rapid adaptation to the demands of the healthcare environment [1,2,3,4].

Career progression and structured professional pathways play a crucial role in enhancing job satisfaction among nursing professionals by providing opportunities for growth, recognition, and development [5,6]. For ECNs, these pathways are particularly important in building the skills and confidence necessary for long-term career success. However, despite the recognised benefits of career progression, research indicates that ECNs often encounter barriers early in their careers [7,8]. The absence of structured training programs tailored for ECNs can hinder their professional growth, as they miss out on learning opportunities crucial for developing expertise in their field [7,8]. While mentorship, support systems, and continuous professional development have been highlighted as critical in facilitating this transition, the long-term career trajectories of ECNs and the factors influencing their professional growth and retention remain underexplored [1,2,3,4].

Furthermore, the professional development and career progression of ECNs are critical not only for individual job satisfaction but also for the sustainability of the healthcare workforce [5]. However, fatigue, often manifesting as emotional exhaustion or compassion fatigue, has significant long-term consequences on job satisfaction and retention rates among ECNs [9,10,11,12]. The relationship between career stress, work stress, and job satisfaction is complex. While career progression can alleviate stress by providing clear goals and pathways, it can also introduce new stressors such as increased workload, heightened expectations, and limited upward mobility [13,14]. Therefore, it is essential to consider the unique challenges faced by ECNs in diverse geographic settings, including both urban and rural areas. Tailoring interventions to meet their specific needs can further enhance their job satisfaction and retention [15].

In addition to the early years of practice challenges, continual learning and further education remain essential in the nursing profession. Obtaining a bachelor’s degree in nursing is just the beginning, with postgraduate study often necessary to achieve career goals in specialised areas such as palliative or acute care [16]. However, the commitment to further study is challenging, with nurses balancing work, family life, and educational pursuits [17]. The need for realistic expectations about the demands of nursing and the importance of lifelong learning is emphasised, as is the necessity for support systems to help nurses manage these competing demands [18].

While nurses demonstrate unwavering commitment and resilience, they also face barriers that need to be addressed to ensure their continued success [4,14,15]. Additionally, creating diverse career opportunities and flexible pathways will further strengthen their dedication and capacity to provide quality care [16]. Within this context, the aim of this study was to interpret and understand the meaning of ECNs’ professional experiences four years after completing their bachelor’s degree in Australia. The key objectives were as follows:To understand how ECNs experienced their professional growth over the four years following graduation.To explore the challenges, limitations, and tensions faced by ECNs in building their professional paths, especially in urban and rural contexts.To analyse the strategies adopted by ECNs to remain professionally adaptable in the face of work demands, personal life, and continuing education expectations.To identify implications for institutional policies and support programs for ECNs’ career progression.

## 2. Materials and Methods

A qualitative study employing a hermeneutic phenomenological approach was conducted to investigate the experiences of ECNs four years after completing their bachelor’s degree education, with the goal of understanding these phenomena [19]. This methodology recognises that language is central to our existence and can be shared to comprehend experiences. According to Gadamer [19], understanding others is achieved through dialogue, following a question-and-answer format. This profound nuanced dialogical method enabled the interpretation of the core structures of participants lived experiences [19].

The study is part of a broader 10-year longitudinal project to explore nursing students’ academic performance and career decision-making while also following them to understand their longevity as nurses within the healthcare workforce [20]. Participants were initially recruited during their final year of study (2018–2020) and were invited for follow-up interviews every two years after graduation. Each participant was interviewed once during 2020–2022 and then again in the 2022–2022 period, depending on their graduation year. This study examines ECNs four years after graduation to allow for a deeper understanding of career trajectories and evolving professional identities.

Given the longitudinal nature of the study, the lead researchers (DT and BP), who were both male, developed a strong rapport with participants over time. While this relationship enhanced trust and openness during interviews, it also introduced the potential for bias. To mitigate this, reflexive practices were embedded throughout the research process. The lead researchers maintained reflexive journals to document assumptions, decisions, and emotional responses. Regular reflexive discussions were held within the research team to critically examine how prior relationships and positionality might influence data interpretation. Independent coding by multiple researchers and peer debriefing further supported the credibility and trustworthiness of the findings.

In addition, peer debriefing involved regular engagement with colleagues and experts to review the research process and challenge interpretations. Reflexivity was further supported by documenting personal reflections and engaging in critical dialogue about how the researchers’ backgrounds and relationships may have shaped the data. These strategies were essential in maintaining analytical rigour and ensuring that the findings remained grounded in participants’ lived experiences.

For additional clarity and transparency, it is acknowledged that this manuscript, along with other articles [21,22], reports on specific aspects of the larger longitudinal research project. This approach allows for a detailed exploration of different facets of the overarching study exploring the career trajectory and profession longevity of ECNs. Findings from this study are reported following the consolidated criteria for reporting qualitative research (COREQ) guidelines [23].

### 2.1. Study Setting and Recruitment

The study was conducted across Victoria, Australia, with participants selected through purposive sampling. Participants included 37 former nursing students who completed a three-year undergraduate nursing degree and participated in an initial study during their final year between 2018 and 2020. Students completed their studies at Federation University Australia, which graduates approximately 300–400 nursing students annually and has campuses across rural, regional, and urban centres throughout Victoria [24]. At that time, all students agreed to be contacted via email 48 months after graduation for an interview. Interviews were conducted between 2022 and 2024. Of those initially interested, 25 agreed to be interviewed, with the remainder either not responding or being too busy to be interviewed. The sample included a diverse group of ECNs in terms of demographics, work settings, and experiences. This diversity helped ensure that a wide range of perspectives was captured, contributing to the robustness of the findings.

### 2.2. Data Collection

Semi-structured interviews were used for data collection. An interview framework was developed and piloted prior to the formal interviews to ensure the clarity, relevance, and comprehensiveness of the questions [22] (Appendix A). The interview framework was adapted slightly over time to reflect participants’ progression in their careers, with additional prompts exploring leadership roles, postgraduate study, and work–life balance. The framework included several standardised questions covering topics such as employment history since graduation, nursing experiences, professional challenges, and positive aspects of nursing. Interviews began with an open question about each ECN’s experiences over the past 12 months, followed by questions about the previous four years.

Data collection took place between April and June each year from 2022 to 2024, with the timing of the four-year follow-up interview depending on the participant’s graduation date. One researcher (D.T.) specialising in the health workforce conducted the interviews via telephone or videoconferencing (MS Teams), which were recorded. Each interview lasted between 20 and 60 min, with most (*n* = 24) exceeding 45 min. Field notes were taken during and after the sessions with the participants’ consent.

Thematic saturation was assessed iteratively throughout the three-year data collection period. After 25 interviews, the research team observed that no new codes or themes were emerging, and the data had become repetitive across participants. This decision was reached through consensus among the lead researchers (DT and BP) and was supported by the diversity of the sample and the depth of the interviews, most of which exceeded 45 min. While Creswell [25] provides a general guideline for sample size in phenomenological studies, our determination of saturation was based on both the richness of the data and the absence of new conceptual insights during the final stages of analysis.

It must be noted that several challenges arose during the data collection process. Coordinating interviews with participants who had varied work schedules and personal commitments demanded flexibility and careful planning. Conducting interviews via videoconferencing sometimes resulted in technical issues, such as poor internet connections or software glitches, which interrupted the conversation flow.

### 2.3. Data Analysis

Interviews were transcribed into Microsoft Word or through videoconferencing technology. The data were checked and cleaned by the two lead researchers (DT and BP). Before analysis, data were labelled based on the year and order of interviews (e.g., Participant (P) 6, 2021; P10, 2022). Thematic analysis was used to identify themes within the data [26], guided by the principles of Gadamerian hermeneutics to ensure the analysis remained rooted in the dialogical process. Initial coding was undertaken independently by the lead researchers (DT and BP), who then met to discuss and refine the codes. Themes were reviewed collaboratively by the larger team to ensure they accurately represented the data. A thematic map was created to visualise the relationships between themes.

Overall, researchers followed the six phases of thematic analysis, starting with familiarisation and immersing themselves in the data by reading and re-reading interview transcripts to deeply understand the content. This immersion enabled an understanding that was continuously refined through engagement with the text [19]. Next, initial codes were generated by assigning meanings to each data set and grouping significant quotes from the interviews. This systematic coding of interesting features across the entire data set helped collate relevant data for each code. The coding process of each code represented a moment of understanding achieved through the interplay of questions and answers. The initial codes were then categorised into potential themes by five researchers (TT, LR, DS, JE, and DT), who independently sorted the different codes and collated all relevant coded data extracts within the identified themes.

Themes were further refined through collaborative discussions and consensus within the research team, ensuring they aligned with the coded extracts and the entire dataset and generating a thematic map of the analysis. Themes were then defined and named to capture their essence, using participant excerpts to enhance confirmability. This ongoing analysis was viewed as a continuation of the hermeneutic dialogue, where themes were not only identified but also interpreted within the context of the participants’ lived experiences. Finally, the report was written by selecting key examples, conducting final analyses of selected extracts, relating the analyses back to the research questions and literature, and producing a comprehensive report of the analysis. This systematic approach to data analysis was chosen for its inductive nature, meaning it did not rely on findings from prior literature or earlier phases of data collection within the same longitudinal study. While participants may have referenced past experiences, the analysis focused on the current dataset, allowing themes to emerge organically and minimising researcher bias, thereby enhancing the trustworthiness of the findings [26].

In addition, as this study employed a hermeneutic phenomenological approach, the analysis was guided by the understanding that meaning is co-constructed through dialogue and interpretation [19]. While the results primarily present descriptive themes supported by participant quotes, some interpretive commentary is included to contextualise the data and preserve the richness of participants’ lived experiences. This approach is consistent with Gadamerian hermeneutics, which emphasises the fusion of horizons between researcher and participant [19]. Broader theoretical interpretations and implications are then reserved for the discussion.

To ensure the rigour of the study, several methods were employed, which included peer debriefing, reflexivity, and member checking. Peer debriefing involved regular discussions among lead researchers (DT and BP), with colleagues and experts included to review and critique the research process, identifying potential biases and areas for further exploration. This refined the analysis and interpretations, ensuring robust findings. The lead research team, experienced in nursing education and health workforce research, maintained reflexive journals to document thoughts, decisions, and potential biases. Regular reflexive discussions and feedback from peers and mentors helped manage biases and ensure balanced analysis.

Member checking involved sharing interview transcripts with participants for feedback, allowing them to confirm the accuracy of the researchers’ understanding, provide additional context, and refine interpretations. Although not always necessary with Gadamerian hermeneutics, member checking strengthened the authenticity of the findings and ensured participants’ voices were represented [27]. Despite no additional comments from participants, this approach fostered a collaborative dialogue and shared understanding of the data [26].

### 2.4. Ethical Considerations

Ethical approval was granted by the Federation University Human Research Ethics Committee (Approval #18-017). This study adhered to the ethical principles of the Declaration of Helsinki and followed all relevant guidelines and regulations. Written informed consent was obtained from each participant at the start of the longitudinal study, with additional written consent sought before each interview.

## 3. Results

Among the participants, a large proportion were female (*n* = 19), aged between 30 and 49 years (*n* = 15), and largely employed in acute hospital settings (*n* = 19), with the remainder working in community and aged care (*n* = 4), while some (*n* = 2) were unable to work due to illness or had left the nursing profession. The majority were working part-time (*n* = 18). The areas in which participants were working varied and included specialties such as emergency (*n* = 2), mental health (*n* = 2), maternity or neonatal intensive care (*n* = 3), and medical or surgical wards (*n* = 6). Participants were geographically diverse, with just under half (*n* = 11) were working in metropolitan or urban areas, while the remaining participants were working in regional or rural areas (Table 1).

Four themes emerged among participants, which encompassed professional growth and unwavering commitment, ongoing professional adaptability, feeling stuck with limited choices, and continual learning amid career challenges and personal life demands (Figure 1). Each of these themes is discussed in detail to provide a comprehensive understanding of the participants’ experiences and perspectives.

### 3.1. Professional Growth and Unwavering Commitment

The participants highlighted several areas of growth that had occurred over their four years in the nursing profession. For example, one nurse stated, “*I initially applied for clinical [role]… the boss called me and said, look, I’d like you to go for clinical lead… it worked relatively well*” (P11, 2023). Another indicated the following:


*My first job… I was the in-charge nurse. I looked after the dementia ward… 60 residents in total… I learned really quickly… I ha[d] to make these decisions so quickly… I can’t muck around.*
(P3, 2022)

Participants revealed a desire for career advancement, with nurses seeking opportunities in areas such as emergency nursing and intensive care, as the following statement indicates:


*I wanted to be in a bigger hospital so I could further [my] career in emergency and then I applied for the postgrad position… I was lucky enough to get a position that shared my work between emergency and started me working in intensive care, which is new for me.*
(P15, 2023)

In addition to considerable growth and development in leadership roles, there was a strong commitment to certain specialities. For some ECNs, aged care was viewed as a long-term career rather than a stepping stone to another specialty. One nurse expressed that they “*committed to a year*” (P14, 2023) in aged care, later adding, “*I’m not just using it as a stepping stone to go somewhere else*” (P14, 2023). This dedication to aged care reflects the nurses’ desire to build deeper expertise and foster the continuity of care rather than merely moving through specialties.

Further, there was an interest in nursing rurally among participants, with nurses expressing their appreciation for the unique opportunities working rurally offers. One nurse shared their experience: “*So I’ve gone from a city environment to the country, and there was no way I would have got this opportunity in the city*” (P14, 2023). The move to a rural setting provided a chance for professional growth through the diverse range of skills required in a rural healthcare environment. Nurses in rural areas often take on a variety of roles that allow them to expand their clinical expertise, as they are frequently required to manage multiple specialties and handle a wider scope of practice than they might in a more specialised metropolitan setting.

This same ECN further explained later, “So, I think rural nursing is excellent. In terms of being able to advance yourself and actually, staying put too, and having that loyalty as well” (P14, 2023). This loyalty, particularly in rural nursing, is crucial. It represents a dedication to the community and the patients who often rely on familiar faces for continuity of care. The sense of commitment to staying and building long-term relationships highlights the unique bond that forms between rural nurses and the people they serve.

Commitment is a key aspect of rural nursing, where nurses take on greater responsibility and develop their skills in challenging environments. One nurse reflected the fooling:


*When you can show that you’re committed to your job and that you don’t have intentions of just ditching when it gets too hard, I think that reflects a lot on you as well. I think that the people around you can see that you’re committed… I’m definitely not going anywhere.*
(P16, 2023)

This commitment was a testament to the enduring relationships and trust that develop within these communities, where the nurses’ dedication is recognised and valued. However, the challenges of limited resources compared to those in metropolitan hospitals were noted, suggesting a need for better support and resources in rural settings. One nurse stated, “*There was a time where we used to have a room [accommodation] upstairs… but since, it has been closed down*” (P16, 2023). They later explained a lack of accommodation and travel difficulties has led some nurses to withdraw from graduate programs.


*We actually had one grad pull out earlier this year because they couldn’t get accommodation. So, I think the desire was there to do the graduate [position].*
(P16, 2023)

In addition to the challenges with accommodation, several nurses discussed the emotional toll of nursing, particularly when dealing with familiar patients. One nurse recounted a difficult experience associated with living and working rurally.


*Because it’s a small town, so you do get to see people that you know. A family friend of ours came in, and [they] went from [being fully awake to unresponsive] on the trolley just waiting to be triaged. She was sent off to [major city], and [they] died.*
(P10, 2022)

It was these profound instances of caring for those within a small, tight-knit community that impacted individual healthcare providers.


*Although I appreciate that we do see people that we know from time to time, to be in that situation where it’s somebody that you know reasonably well is incredibly difficult.*
(P10, 2022)

Amid the challenges, nurses revealed significant avenues for growth within the profession, particularly in rural and aged care settings. While nurses demonstrate unwavering commitment and adaptability, they do encounter challenges that need to be addressed to ensure their continued success. Improving resources in rural areas, offering better accommodation and travel solutions, and providing emotional and mental health support were highlighted as critical to fostering an environment where nurses can thrive. Additionally, creating diverse career opportunities that allow for professional advancement will further strengthen their dedication and capacity to provide quality care.

### 3.2. Ongoing Professional Adaptability

In addition to professional growth and commitment, the nurses depicted a dynamic and evolving journey within the nursing profession. Nurses discussed the transitions within their roles, highlighting that fluidity and adaptability are required. Several nurses indicated that they were still moving from one department to another, such as moving from a ward to the emergency department to find their career pathway. These transitions often stem from personal interest, professional growth opportunities, or staffing needs within the hospital. For example, one nurse indicated the following:


*Emergency is always short, and they asked me if I wanted to go down there and do a shift… I went there and loved it. When a job came up for Emergency in December, I applied and got in.*
(P9, 2022)

Another stated, “*I don’t want to be doing the same thing every day*” (P3, 2022). Others had found their niche and stayed where they were: “*I’m still working in the same ward*” (P4, 2022).

Promotions and role changes are significant milestones for some nurses, reflecting their professional development and recognition. “*Being promoted was, you know, a big stepping stone for me. I just felt so lucky and so blessed to be able to have that position*” (P14, 2023). In addition, others indicated their desires to grow and develop:


*I’m going to be starting up some in charge training… after that I really would like to look into going into emergency mental health.*
(P24, 2024)


*I always wanted to be able to manage a facility. So hopefully one day I might get that opportunity and who knows, you know, with all the opportunities I’ve had so far and where I’m where I’m heading in terms of what I’m learning now, it’s sort of gonna open up the door a little bit more, that’s for sure.*
(P14, 2023)

However, for some other nurses, these advancements in their careers often came with new challenges and responsibilities, requiring nurses to adapt and learn quickly: “*I really put myself out of my comfort zone. But I don’t regret it, I absolutely love it*” (P3, 2022).

Balancing personal life with professional commitments was a recurring theme, with some nurses discussing returning to work after maternity leave, adjusting their work schedules, and juggling multiple roles. “*I’m currently casual but looking to apply for a part time role and will try to move a bit closer [to home]*” (P13, 2023). The nurses’ narratives highlight the flexibility and resilience needed in the nursing profession. For example, one nurse stated, “*I returned to work in the casual part—time pool, after working in Safety and Quality*” (P6, 2022). Another stated, “*I am now full time PhD, so I still work [clinically] one day a week.*” (P4, 2022)

Several nurses expressed a desire for change or a need for new challenges, leading them to seek different roles or specialties within nursing. “*I got itchy feet; it was time to challenge myself*” (P3, 2022). They discussed the excitement and fulfilment they derive from these new experiences, despite the initial discomfort of stepping out of their comfort sones. “*I got sent to ED from Pool. I went there and found I loved it. So, they just kept sending me there*” (P9, 2022). Another, stated “*I kind of fell into it, to be honest with you*” (P20, 2024).

However, in some cases, the challenges of the workplace instigated the need to be adaptable and transition into other areas of nursing. This was clearly articulated when a nurse stated the following:


*I was physically assaulted by a drug affected patient… and had serious injuries that required time off work… It was around the time that that patient assaulted me that the applications opened up for Midwifery and I thought, well, one, now’s a good time as any.*
(P21, 2024)

Similarly, another nurse stated she moved back to her former ward when the specialisation she moved into, neonatal intensive care, was not the right fit, stating, “*I wasn’t feeling that I was as supported as I was when I was in a ward environment… So, I’m back. I’m working in Paediatrics*” (P19, 2024).

Overall, the nurses’ experiences provide some insight into the diverse pathways they both trialled and found within the nursing profession to meet their needs and how they adapted to finding the right fit or needing to change based on individual circumstances or need. Their experiences encapsulate the continuous learning, adaptability, and resilience that characterise a nursing journey.

### 3.3. Feeling Stuck with Limited Choices

Despite some nurses feeling they had the capacity to be adaptable and try new things, other nurses also highlighted a palpable sense of entrapment due to their specialised areas of work. Specialisation entrapment, or the feeling of being stuck in a specific field without the flexibility to transition to other areas, was clearly articulated. Several nurses felt they were stuck, as they felt they could not return to general nursing, particularly for those who specialised in mental health upon graduation. One nurse, when reflecting on what they would tell themselves as a student, said the following:


*I would probably tell myself ‘don’t specialise in mental health’, because once you specialise in mental health, you cannot go back to regular nursing.*
(P13, 2023)

Specialisation was suggested to limit their skills to a specific area, making them feel less confident about their abilities in other nursing domains. The erosion of confidence that comes with prolonged specialisation was apparent when another stated the following:


*It is hard to imagine myself going back into the clinical side of things…I feel like the longer I stay away from… that clinical role, the less confident… in my own capabilities in my own skills and knowledge.*
(P4, 2022)

Another nurse in mental health also added the following:


*I have thought what am I gonna do? If I stepped on to a Med[ical] Surg[ical] ward or something like that… I won’t remember how to do some of these things… sometimes I do feel a bit sad about that you know, I learned all that at Uni[versity] and I’m not sort of doing it every day.*
(P24, 2024)

Finally, specialisation can limit professional careers by narrowing the scope of opportunities available to nurses. The pressure to specialise early in their careers is a recurring theme, with some nurses expressing regret about their choices and the limitations these decisions have imposed on their career trajectories. Others already knew they could not continue in the long term:


*I also am at the point where I want to do more client work rather than sitting in an office and writing policies and procedures… I think I’m a bit too young into my career to be doing that at this stage.*
(P13, 2023)

The nurses felt their specialisation might limit their career progression, such as teaching clinical subjects at a university.


*It potentially may harm my career down the track in terms of I know for example, if I was to go into teaching at the university. I probably will not be able to teach any of the clinical subjects that would be more.*
(P4, 2022)

This limitation can hinder professional growth and development, making it challenging for nurses to advance in their careers: “*There’s no, like, no job will accept you*” (P13, 2023). Overall, the nurses highlighted the need for more flexible career paths within nursing to prevent feelings of being stuck in one area. The nurses felt less capable of working in other nursing areas, which contributed to the feeling of being stuck: “*You can’t even go on the casual [pool] at your local hospital because you don’t know how to do your infusions and stoma care*” (P13, 2023).

It was suggested there was a lack of programs to facilitate a transition from a specialised nursing area, such as mental health, back to ward nursing, exacerbating the issue of feeling stuck. One nurse stated, “*There are plenty of programs to go from regular nursing into mental health, but no, none to get back the other way*” (P13, 2023). This lack of flexibility made some nurses feel stuck and unable to pursue other interests within nursing, such as emergency department roles. For example, one ECN stated, “*I feel like the longer I stay away from clinical roles, the less confident I am I going back… it potentially may harm my career down the track*” (P5, 2022). Additionally, this lack of support makes it difficult for nurses to regain the necessary skills and confidence in other nursing areas, further limiting their career mobility: “*I feel like I have less confidence in my own capabilities in my own skills and knowledge because it has been*” (P4, 2022). Conversely, another nurse, who was experiencing the same issues, returned to ward nursing and was surprised.


*So, I went back and did a shift, and I was so nervous. I was like, I’m not gonna remember how to do bladder wash out… then I went, and it was just like muscle memory… going back into the ward, I was like, ‘oh, this is so nice.’*
(P7, 2022)

In addition to feeling stuck due to career choices and limited opportunities to move into different roles making the nurses question their career choices, other nurses indicated they also felt stuck in their careers due to financial commitments. For example, a nurse stated, “*Many people had high financial commitments and what else are they gonna do? You know they’re going well*” (P17, 2023). Nurses were indicating they had high financial commitments, and they felt trapped in their current role due to a financial obligation. However, others moved to more local yet limited opportunities, “*where the rent was affordable*” (P23, 2024).

Such limitations and less flexibility may lead to stress and burnout, as nurses may feel they have no sense of control and no other options but to continue in their current job despite any dissatisfaction or exhaustion they might be experiencing. Furthermore, another nurse said when discussing their financial commitment and employment as a nurse to meet that obligation, “*what else are they gonna do*?” (P17, 2023). This implied a sense of hopelessness and lack of alterative career pathways, which led them to question whether they made the right career choice.

It was indicated that many factors centred on feeling stuck, along with burnout often causing the nurses to question their career choices, wondering if they made the right decision and if the career is worth the toll it takes on their wellbeing.


*I worked nothing under 90 odd hours a fortnight… I was on permanent nights for the last seven months of [emergency department] because nobody would work nights… I was living to work not working to live.*
(P21, 2024)

In addition to the challenges of the toll nursing takes and feeling stuck, it was indicated that the university did not really prepare the nurses for these challenges. One participant’s comments encapsulate what others were verbalising:


*There was never any indication of just how hard nursing was gonna be and how taxing it would be and how, like how much you’re gonna second guess yourself and how much you’re gonna, you know, I think, have I done the right thing? Is this career worth it?*
(P13, 2023)

The nurses express a need for more realistic expectations about the challenges of nursing to be communicated during their education as students.

### 3.4. Continual Learning Amid Career Challenges and Personal Life Demands

Beyond feeling stuck, nurses highlighted further education as a way to move into other areas of nursing; however, there was realisation that more study was required and that gaining a bachelor’s degree was merely the beginning.

*I kind of was under the illusion when I was a student that once I have my Bachelor’s degree, I’m good to go and I don’t need to do any further study unless I want to, you know, specialise. But that’s really not the case in nursing. You have to do extra study no matter where you end up*. (P13, 2023)

There was insight that postgraduate study could be used to realise career goals such as working in areas of interest, including palliative or acute care.


*So, I went into that with maybe sort of the idea of all these subjects that I’ve done can transfer over if I decide to pursue my postgraduate in, say, critical care nursing.*
(P9, 2022)

Likewise, there was recognition that career goals may not be achieved without further study: “*I will have to do it because everywhere else… I… wanna work is requiring a postgrad[uate degree]*” (P13, 2023). Conversely, other participants found the notion of further study inviting because it means working in an environment they enjoy.

*I’m liking where I’m sitting within the organisation and be happy to do further study to maintain a role within them. So, if that’s something that they are looking for, definitely open to that*. (P6, 2022)

Others recognised that further study was an expectation and was a paramount element to meet the employability expectation, with one participant recalling to themself, “*I’ve gotta get this study done before I can get the job. But already got the job because… you had to show you were gonna do the study*” (DL22). Another said, “*There were opportunities to undertake training, so my work has paid for every single part of training I’ve wanted to do*” (P22, 2024).

However, the reality and prospect of further study was hard for some participants to embrace. The commitment to study was seen to intrude and provide unwanted pressure in all areas of life. Some participants wanted to focus on areas of their life outside of work. Additional study caused a sense of conflict.


*I actually wanted to do my postgrad certificate in palliative care. I wanted to do I still wanna do it, but I really can’t be bothered. I wanna come home and I wanna get in my garden. I wanna brush my horse. I don’t wanna commit to having to study. I don’t wanna have to have that pressure.*
(P7, 2022)

For some, the reality was that it was hard to juggle work, family life, and study, while such personal sacrifices were already experienced and had impacted personal lives. “*I’ve got exciting things happening and I missed out on things for such a long time while I was studying. I don’t wanna do at the moment*” (P7, 2022). This sentiment was also reflected by another student who stated that studying had impacted them:


*It’s… not actually participating in family life that makes it more difficult, even if they’re not putting pressure on you, it’s still your own internal pressure that you’re putting on yourself. It’s nice to actually get a weekend off, but [feeling] I should have been studying on most weekends, too.*
(P4, 2022)

Postgraduate study is not only a commitment that requires sacrifice; it can also be challenging for other reasons. “*I have ADHD, so study is very hard for me*” (P13, 2023). Motivation to study is difficult to find when the perception is that it is not yet essential.


*Technically I’m still… enrolled and everything like that. Have I put any work towards that? At the moment, no, because it’s like it hasn’t actually been needed for my job at this point.*
(P13, 2023)

Other priorities were considered essential, and it was unclear where postgraduate study fit into this reality. Some participants indicated that the timing may not always be right and may need to be adjusted.


*I made the decision not to go on and do postgraduate studies straight away just because of the demands of my family and that they are my first value and priority, and I had spoken to my nurse unit manager about that. Said I’m gonna wait one more year. Learning wise, I think I just need to sit with it and take a breath for a while.*
(P17, 2023)

Participants were saying “not yet” rather than “not ever”, with the intention to study still prevailing for many. When the time was right to study, several participants noted that before embarking on further study, preparation was required. Preparation included physical considerations such as location: “*Once we settle down, in the western Suburbs, I’m looking at enrolling into a PhD and research*” (P1, 2022). Adjustments to working hours were also necessary to accommodate study requirements. “*I was working 0.6* [part time]… *and I was working [elswhere]… one day a week and I also commenced a Grad[uate] Cert[ificate] in Tertiary Education*” (P4, 2022).

Mental preparation was also required. One participant mentioned, “*I do want to do a post grad, but mentally. Not yet. No, because obviously I’ve just been whacked around for a little bit*” (P16, 2023) when speaking of their work and home commitment. Further, another shared similar feeling when they stated the following:


*Just the way things go with how busy we got. I just lost a bit of the motivation. It’s just I know it’s going to survival mode, and you go to work, you deal with what you have to and then you go home, and you try and forget about everything that you’ve, you know you’ve had.*
(P10, 2022)

For those who chose to pursue postgraduate study, the recognition of its value to future clinical practice was appreciated. “*I’m really grateful that I decided to do my postgrad studies as well*” (P3, 2022). Overall, participants revealed no singular consensus on when or how nurses may decide to pursue postgraduate studies. However, what was evident was the complexity and consideration that must be given to enabling nurses to continue their lifelong learning journey while managing the perceived barriers.

## 4. Discussion

This study explores the impact of various work-related factors on the professional growth, adaptability, challenges, and lifelong learning of ECNs during their first four years in the profession. The findings indicated that, despite the challenges they encounter—including emotional fatigue, and resource limitations—ECNs largely perceive nursing as a meaningful and fulfilling career. However, their professional journey is shaped by a delicate balance between career aspirations and personal wellbeing. Many nurses prioritise maintaining a sustainable work–life balance, emphasising the need for structured support systems that allow them to dedicate time to family, personal health, and professional growth without experiencing burnout. Furthermore, career advancement opportunities, ongoing learning, and financial stability emerged as crucial factors influencing ECNs’ long-term commitment to the profession. The evolving expectations within the nursing workforce, particularly concerning career flexibility, postgraduate education, and workplace support, highlight the need for systemic strategies to ensure both professional satisfaction and workforce retention of ECNs.

Nurses’ career decisions and goals are highly individualised, shaped by a complex interplay of personal interests, preferences, priorities, and external factors such as workplace conditions, career opportunities, and proximity to family and social support networks [28,29]. While some ECNs prioritise career progression in specialised fields, others make strategic choices based on lifestyle considerations, job security, or geographical convenience. This observation aligns with London’s [30] theory of career motivation, which posits that career decisions and behaviours are influenced by both intrinsic characteristics (e.g., personal values, aspirations, and competencies) and extrinsic situational conditions. Our findings also resonate with the sustainable careers framework proposed by De Vos et al. [31], which highlights the significance of adaptability and continuous negotiation between individual career aspirations, employer expectations, and broader system factors. The extent to which ECNs can exercise agency in shaping their career paths is largely contingent on the structural opportunities and constraints within their work environments, which highlights the need for flexible career pathways that support professional growth and long-term job satisfaction.

The results corroborate previous studies [15], which have shown that ECNs are deeply committed to the nursing profession, demonstrating a strong sense of purpose and dedication to patient care. However, while professional commitment remains high, ECNs increasingly seek a holistic perspective on work, striving to establish boundaries that foster a healthy work–life balance and sustain long-term engagement [32,33]. Many ECNs reported actively setting boundaries to prevent emotional exhaustion and burnout, particularly given the demanding nature of nursing work. Strategies such as reducing overtime, transitioning to part-time roles, and selecting work environments with supportive team cultures were common approaches used to maintain wellbeing. Some ECNs opted for career pathways that provided greater flexibility, enabling them to exert more control over their schedules and workloads. This shift toward prioritising sustainable work practices reflects broader trends in workforce retention, where nurses are increasingly making career decisions that balance professional aspirations with personal wellbeing [34].

This study revealed that employability expectations are significant triggers for pursuing further education, often shaping ECNs’ career trajectories and professional development plans. Many ECNs reported that while postgraduate qualifications were increasingly viewed as essential for career advancement, the decision to undertake further study was complex and influenced by multiple factors. The financial burden of additional education was a significant concern. Beyond financial implications, many ECNs expressed concerns about the impact of study commitments on their personal lives, highlighting the challenge of balancing professional development with family responsibilities, social engagement, and overall wellbeing. The decision to pursue further education was often weighed against life-stage considerations, with some ECNs delaying study due to changes in personal life, such as starting a family [35]. Workplace expectations also played a role, as some ECNs felt pressured to obtain further qualifications to remain competitive in specialised fields. These findings suggest that while lifelong learning is essential in nursing, greater financial assistance, flexible study options, and employer-sponsored training may be needed to improve access to postgraduate education for ECNs.

Much of the learning that nurses are expected to engage in is workplace-driven, focusing on contextual knowledge, such as new technologies or emerging treatments [36]. Employers often require nurses to obtain various workplace-related certificates and credentials, many of which emphasise the instrumental aspects of nursing care rather than broader professional development [37]. While these certifications enhance capacity and capability in specific clinical areas, they can also create an ongoing expectation for nurses to independently manage their professional learning without sufficient institutional support. The emphasis of lifelong learning is increasingly framed as a necessity for career sustainability, which requires nurses to continuously upskill in order to remain employable and flexible in a rapidly evolving healthcare environment [36]. However, this expectation can place additional pressure on ECNs, particularly those already struggling with workload demands and financial constraints.

The realities of the workplace reveal a persistent gap between the expectation for ongoing learning and the financial and structural support available to nurses. While ongoing professional development is essential for career progression, many nurses are not financially supported for ongoing learning [35]. This creates a paradox where nurses must continuously invest in their education to remain competitive in an increasingly demanding job market, despite the lack of financial support [38]. The financial burden is further compounded by the need to balance study commitments with full-time employment, leading to increased stress and work–life conflicts. In addition, the pressure to meet professional standards and the desire for career advancement often clashes with nurses’ desire for career flexibility, personal wellbeing, and life satisfaction. Many ECNs struggle to navigate these competing demands, with some delaying further education or reconsidering their career pathways to avoid additional strain.

### 4.1. Implications

Policy and institutional changes are needed to ensure professional learning becomes a shared responsibility among nurses, employers, and educational institutions. Policymakers and healthcare leaders may consider a more structured and collaborative approach to supporting ECNs by developing career pathways, improving workplace conditions, and expanding access to continuous learning. In addition, strengthened career transition programmes will be helpful in enhancing professional mobility and reducing career stagnation. Organisational support and adequate staffing levels may also play a crucial role in improving job satisfaction and retaining ECNs. Mentorship initiatives and structured career development frameworks may further assist ECNs in navigating their long-term career trajectories.

Overall, the study highlights several targeted strategies regarding how to operationalise these recommendations. Hospitals and health services may consider implementing structured, flexible career pathways that allow ECNs to explore diverse clinical areas without the risk of career entrapment. This includes the development of re-entry programs for nurses wishing to transition between specialties, particularly from mental health back to general nursing. Universities may benefit students by embedding more realistic insights regarding the nursing profession within undergraduate curricula, including the emotional demands, career adaptability, and the necessity of lifelong learning. Health agencies and policymakers may consider expanding financial and structural support for postgraduate education, such as subsidised tuition, protected study time, and employer-sponsored training. These measures may alleviate the financial and emotional burden associated with further study and enable ECNs to pursue advanced roles without compromising personal wellbeing, thereby contributing to a more resilient and adaptable nursing workforce.

### 4.2. Limitations

This study was conducted within Australian healthcare context. Although participants from diverse backgrounds were included and detailed descriptions of the research context and findings were provided, along with sufficient and representative extracts (i.e., vivid quotes from participants) to help readers assess transferability, the findings may not be fully generalisable to other healthcare systems with different workforce structures and policies. Qualitative methods, while effective in capturing nuanced experiences, may limit the broader applicability of findings due to sample size constraints. Further research employing mixed-methods approaches could further enhance understanding by incorporating larger, more representative samples alongside in-depth qualitative analyses. Furthermore, the impact of external factors, such as organisational policies or broader health system changes, was not explicitly explored in this study, warranting further investigation into how these structural elements shape the ECNs’ career trajectories.

It is vital to acknowledge that the graduation and early career experiences of the participants coincided with the COVID-19 pandemic (2020–2022) [39]. While the interviews were conducted between 2022 and 2024, participants graduated between 2018 and 2020, and the pandemic formed a significant backdrop to their transition into practice. Despite this overlap, participants did not explicitly reference the pandemic as a major influence on their career decisions, transitions, or perceptions during the interviews. This absence may reflect a shift in focus toward more enduring structural and personal challenges encountered in the profession, or it may suggest that the pandemic’s impact had been internalised or normalised by the time of the interviews. Nevertheless, the broader context of the pandemic may have likely shaped the healthcare environment in which these ECNs began their practice, and this should be considered when interpreting the findings.

### 4.3. Future Research Opportunities

While this study provides valuable insights into the lived experiences of ECNs four years after graduation, further research is needed to build on these findings. Future studies are needed to explore the long-term impacts of structured career transition programmes on nurse retention and job satisfaction, particularly in rural and remote settings. Quantitative or mixed-methods research could assess the effectiveness of re-entry pathways for nurses transitioning between specialties, specifically from mental health back to medical or surgical nursing. Additionally, longitudinal studies examining the outcomes of employer-supported workplace thriving initiatives or postgraduate education with a focus on career progression, burnout reduction, and workforce sustainability would provide critical evidence to inform policy. Investigating the role of institutional mentorship programs and their influence on ECNs’ career adaptability and resilience would also be beneficial. Finally, comparative studies across different healthcare systems could help identify best practices in supporting ECNs internationally [40].

## 5. Conclusions

ECNs face a complex career landscape, balancing aspirations with workplace constraints and personal commitments. Many ECNs remain committed to nursing, yet career entrapment, job stress, and financial barriers to further education create significant hurdles. Workforce reforms are crucial for strengthening career flexibility, enhancing institutional support for lifelong learning, and improving job conditions to boost retention. Targeted policy changes could also help sustain a resilient nursing workforce. Future research is warranted to examine career mobility trends, structured transition programs, and financial support mechanisms to build a more adaptable and engaged nursing workforce.

## Figures and Tables

**Figure 1 nursrep-15-00214-f001:**
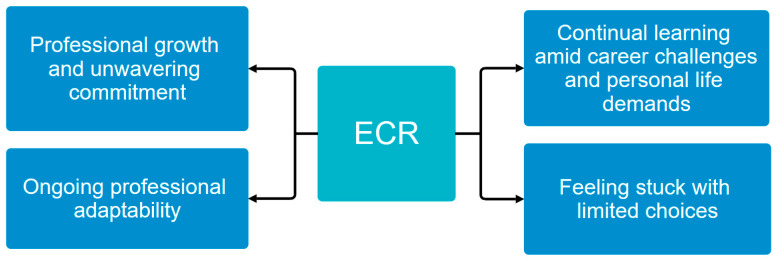
Key themes among participants’ lived experience.

**Table 1 nursrep-15-00214-t001:** Participant characteristics.

Demographic Information	*n* = 25	(%)
Gender		
Female	19	76
Male	6	24
Age group		
20–29	8	32
30–39	9	36
40–49	6	24
50–59	2	8
Current workplace		
Hospital		
Aged care	19	76
Primary healthcare/community	1	4
Not currently working clinically	1	4
	4	16
Current workplace—geography		
Metropolitan/urban		
Rural/regional	11	44
Not applicable	10	40
	4	16
Hours currently working (full-time equivalent hours)		
Full time	1	4
Part time	18	72
Casual	2	8
Not applicable	4	16

## Data Availability

The data presented in this study are available on request from the corresponding author due to requiring ethical clearance to share the data.

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
