# Peer review of "Embracing Growth, Adaptability, Challenges, and Lifelong Learning: A Qualitative Study Examining the Lived Experience of Early Career Nurses"

_nursrep, 2025, doi:10.3390/nursrep15060214_

Round 1

Reviewer 1 Report

Comments and Suggestions for Authors

Thank you for the opportunity to review this manuscript. This details the analysis of early career nurses at the 4-year mark following prelicensure nursing education. It is part of a longitudinal study and is presented as stand-alone data. Generally, the manuscript flows well and information is provided to assist the reader to understand the study and replicate if desired. I have four primary areas that I think need clarification. First, on line 177 is the phrase “does not consider previous research outcomes”. Are these research outcomes from other studies? Or are you referring to interviews with these same participants and research team members? If the latter, unless NO participant referred to or compared/contrasted from previous data-gathering in the longitudinal study, I don’t think you can make this statement? Please clarify. Secondly, when I examine the graduation dates and the dates of the current interviews, I notice that they coincide with the COVID pandemic. Did no participant mention that the pandemic may have influenced their perceptions related to early career choices, transition into practice, etc.?  You may choose to include this in the discussion rather than with the themes, but it should be mentioned in the absence or presence. Thirdly, while the results were well written and supported with quotes, at times the narrative tended to lean into discussion territory. Perhaps re-titling or providing a very short explanation at the beginning of the results section would help. Finally, a bit more information on the longitudinal study would help the reader to know how to use the results. For example, how often are the participants interviewed? Did the questions change from one interview to another to accommodate progression in the profession? How long will you follow these participants?

Author Response

Reviewer #1

Comment

Response

Thank you for the opportunity to review this manuscript. This details the analysis of early career nurses at the 4-year mark following prelicensure nursing education. It is part of a longitudinal study and is presented as stand-alone data. Generally, the manuscript flows well and information is provided to assist the reader to understand the study and replicate if desired. I have four primary areas that I think need clarification.

Thank you, we hope we have addressed all comments and questions raised.

First, on line 177 is the phrase “does not consider previous research outcomes”. Are these research outcomes from other studies? Or are you referring to interviews with these same participants and research team members? If the latter, unless NO participant referred to or compared/contrasted from previous data-gathering in the longitudinal study, I don’t think you can make this statement? Please clarify

Thank you for your observation. To clarify, our intent was to convey that the analysis was conducted inductively, without being influenced by prior literature or earlier phases of data collection within the same longitudinal study. While we acknowledge that participants may have referenced past experiences, the coding and theme development centred solely on the current dataset. We have revised the wording in the manuscript to better reflect this distinction and avoid potential misinterpretation. We hope this addresses your concern and strengthens the transparency of our methodological approach.

(Line 206-210)

Secondly, when I examine the graduation dates and the dates of the current interviews, I notice that they coincide with the COVID pandemic. Did no participant mention that the pandemic may have influenced their perceptions related to early career choices, transition into practice, etc.?  You may choose to include this in the discussion rather than with the themes, but it should be mentioned in the absence or presence.

Thank you for your observation. We acknowledge this important contextual factor and have now addressed it in the revised manuscript. Specifically, we have added a paragraph in the Limitations section to note that while the pandemic coincided with the early career period of participants, it was not explicitly mentioned by them as a significant influence on their career decisions or transitions. We also reflect on possible reasons for this absence and its implications for interpreting the findings.

(Line 670-680)

Thirdly, while the results were well written and supported with quotes, at times the narrative tended to lean into discussion territory. Perhaps re-titling or providing a very short explanation at the beginning of the results section would help.

Thank you for your thoughtful feedback regarding the Results section. We appreciate your observation that some parts of the narrative may appear interpretive in nature. To clarify this, we have added a short paragraph in the data analysis section to explain the approach used. As this study is guided by hermeneutic phenomenology, our aim is not only to describe participants’ experiences but also to interpret the meanings embedded in their narratives. This methodology recognises that understanding is co-constructed through dialogue and interpretation. Therefore, while the Results section primarily presents descriptive themes supported by participant quotes, some interpretive commentary is included to provide context and preserve the richness of the lived experiences shared. To maintain clarity, we have ensured that broader theoretical analysis and implications are reserved for the Discussion section. We hope this explanation helps clarify the structure and intent of the Results section and aligns with the expectations of qualitative research reporting.

(Line 211-218)

Finally, a bit more information on the longitudinal study would help the reader to know how to use the results. For example, how often are the participants interviewed? Did the questions change from one interview to another to accommodate progression in the profession? How long will you follow these participants?

Thank you for this question and suggestion. We have now expanded the Materials and Methods section to provide additional detail about the longitudinal design of the study. We also note that while the core interview framework remained consistent, it was adapted slightly over time to reflect participants’ career progression, with additional prompts exploring leadership roles, postgraduate study, and work-life balance. We hope this has enhanced transparency and help readers better understand the scope and application of the findings.

(Line 104-111)

Reviewer 2 Report

Comments and Suggestions for Authors

Dear Authors,
The manuscript is well-written, cohesive, and robust. The methodological approach is thoroughly described in accordance with the COREQ guidelines. The results appropriately address the research objective, and the discussion is broad and relevant to the topic. Below, I provide a few suggestions to further enhance the quality of the manuscript:

  • Remove the descriptors that are not indexed in MeSH, such as Professional Development, and include the following descriptors instead:

    • Nurse’s Role

    • Professional Practice

    • Education, Nursing, Continuing

    • Staff Development

    • Policy Making

  • The descriptor career mobility does not appear to be appropriate for this study.

Introduction

  • Please insert a bibliographic reference for the statement presented in lines 80–81.

  • I suggest removing the following sentence, which likely reflects a local context specific to Australia:
    “Improving resources in rural areas, offering better accommodation and travel solutions, and providing emotional and mental health support are critical to fostering an environment where nurses can thrive [17].”

General Objective

  • I suggest rephrasing the general objective as follows:
    “To provide a comprehensive understanding of the professional experiences of ECNs four years after completing their bachelor's degree in Australia.”

  • Please include the same objective in the abstract.

Methods

  • Under Study setting and recruitment, please include the following information: At which university did the students complete their undergraduate degree? Is it located in the capital or a rural area? How many students graduate annually?

  • Line 125 – Is it possible to make the interview framework available as supplementary material?

  • Lines 144–145 – Were these interviews re-conducted?

  • Lines 147–148 – Please specify which videoconferencing technology was used.

Results

  • Review the formatting of Table 1.

  • Line 216 – Consider replacing home demands with personal life demands. If this change is made, please update the abstract accordingly.

  • Check the formatting of the participants’ quotes – all quotes should be in italics

Author Response

Reviewer #2

Comment

Response

The manuscript is well-written, cohesive, and robust. The methodological approach is thoroughly described in accordance with the COREQ guidelines. The results appropriately address the research objective, and the discussion is broad and relevant to the topic. Below, I provide a few suggestions to further enhance the quality of the manuscript:

Thank you, we hope we have addressed all comments and questions raised.

Remove the descriptors that are not indexed in MeSH, such as Professional Development, and include the following descriptors instead:

Nurse’s Role

Professional Practice

Education, Nursing, Continuing

Staff Development

Policy Making

The descriptor career mobility does not appear to be appropriate for this study.

Thank you for this suggestion, we have now revised these descriptors as suggested.

(Abstract)

Introduction

Please insert a bibliographic reference for the statement presented in lines 80–81.

I suggest removing the following sentence, which likely reflects a local context specific to Australia:

“Improving resources in rural areas, offering better accommodation and travel solutions, and providing emotional and mental health support are critical to fostering an environment where nurses can thrive [17].”

Thank you, we have undertaken the following:

Added bibliographic reference for the statement presented in lines 80–81

(Line 82)

We have removed the statement as suggested.

(Removed)

General Objective

I suggest rephrasing the general objective as follows:

“To provide a comprehensive understanding of the professional experiences of ECNs four years after completing their bachelor's degree in Australia.”

Please include the same objective in the abstract.

Thank you, we have undertaken the following:

We have rephased the objectives as suggested.

(Line 85-94)

We have included the same in the abstract as directed.

Methods

Under Study setting and recruitment, please include the following information: At which university did the students complete their undergraduate degree? Is it located in the capital or a rural area? How many students graduate annually?

Line 125 – Is it possible to make the interview framework available as supplementary material?

Lines 144–145 – Were these interviews re-conducted?

Lines 147–148 – Please specify which videoconferencing technology was used.

We have included the specific data regarding the higher education institution within the methods section as requested

(Line 137-139)

We have included this as an appendix of the paper.

(Appendix A)

We have provided additional details regarding the interviews (Line 104-1011)

We have added the details of the videoconferencing technology

(Line 160)

Results

Review the formatting of Table 1.

Line 216 – Consider replacing home demands with personal life demands. If this change is made, please update the abstract accordingly.

Check the formatting of the participants’ quotes – all quotes should be in italics

Thank you, we have undertaken the following:

We have revied table 1, which is now in line with the formatting of the journal requirements

(n/a)

We have replaced the wording with the suggested wording throughout the manuscript

(Throughout)

We have made these changes as suggested, thank you

(Throughout)

Reviewer 3 Report

Comments and Suggestions for Authors

Abstract:
In the abstract, the methods section should mention the number of participants in the study.

Introduction:
Since the approach is hermeneutic phenomenology, it would be helpful to explicitly state that the objective is to understand the meaning of ECNs’ experiences over time — something more interpretive than simply "investigating" or "exploring."

I believe the article would benefit, in terms of clarity, if the objectives were more clearly defined. Suggested reformulation of the objectives:

- To understand how early career nurses (ECNs) experienced their professional growth over the four years following graduation.

- To explore the challenges, limitations, and tensions faced by ECNs in building their professional paths, especially in urban and rural contexts.

- To analyze the strategies adopted by ECNs to remain professionally adaptable in the face of work demands, personal life, and continuing education expectations.

- To identify implications for institutional policies and support programs for ECNs’ career progression.

Methodology:
The study mentions that the lead researchers had a "strong relationship" with the participants since graduation (lines 105–106), but it does not discuss potential biases arising from this relationship. A more critical section on reflexivity and the measures taken to mitigate researcher influence on the data should be included (beyond stating that “reflexive journals” were maintained).
The sample consisted of 25 participants, but the study does not explain how thematic saturation was reached, other than a generic citation of Creswell.

Results:
A table or visual figure could be included to synthesize the four main themes and subthemes. This would allow for better visualization of the findings.

Conclusion:

- It should be stated how the results can guide specific interventions in hospitals, universities, or health agencies. In other words, what are the implications for practice? What new insights does this study bring to clinical practice and health policy development?

- More concrete suggestions for future research could be highlighted.

Author Response

Reviewer #3

Comment

Response

Abstract:

In the abstract, the methods section should mention the number of participants in the study.

Thank you, we have moved the number of participants noted in the results to the methods section of the abstract

Introduction:

Since the approach is hermeneutic phenomenology, it would be helpful to explicitly state that the objective is to understand the meaning of ECNs’ experiences over time — something more interpretive than simply "investigating" or "exploring."

I believe the article would benefit, in terms of clarity, if the objectives were more clearly defined. Suggested reformulation of the objectives:

- To understand how early career nurses (ECNs) experienced their professional growth over the four years following graduation.

- To explore the challenges, limitations, and tensions faced by ECNs in building their professional paths, especially in urban and rural contexts.

- To analyze the strategies adopted by ECNs to remain professionally adaptable in the face of work demands, personal life, and continuing education expectations.

- To identify implications for institutional policies and support programs for ECNs’ career progression.

Thank you for this suggestion, we have revised the aim and objectives as suggestion (Abstract and Line 85-94)

Methodology:

The study mentions that the lead researchers had a "strong relationship" with the participants since graduation (lines 105–106), but it does not discuss potential biases arising from this relationship. A more critical section on reflexivity and the measures taken to mitigate researcher influence on the data should be included (beyond stating that “reflexive journals” were maintained).

The sample consisted of 25 participants, but the study does not explain how thematic saturation was reached, other than a generic citation of Creswell.

Thank you for this feedback. We have revised the manuscript to include a more detailed and critical discussion of reflexivity. Specifically, we now acknowledge the potential for bias arising from the strong rapport developed between the lead researchers and participants over the course of the longitudinal study. To address this, we provided more detail regarding the reflexive strategies employed, including the use of reflexive journals, regular team discussions, and peer debriefing to critically examine how researcher positionality may have influenced data interpretation. These measures were taken to enhance transparency and ensure the trustworthiness of the findings. (Line 112-126)

Thank you, we have also expanded our explanation of how thematic saturation was determined.  (Line 163-170)

Results:

A table or visual figure could be included to synthesize the four main themes and subthemes. This would allow for better visualization of the findings.

Thank you for this suggestion. We have now included a visual figure to assist with the synthesis of the four main themes. There were no sub themes.

(Line 258)

Conclusion:

- It should be stated how the results can guide specific interventions in hospitals, universities, or health agencies. In other words, what are the implications for practice? What new insights does this study bring to clinical practice and health policy development?

- More concrete suggestions for future research could be highlighted.

Thank you for these comments and suggestions, we have made some additional changes in line with each.

(Line 643-656)

This has also been added

(Line 681-694)

Round 2

Reviewer 3 Report

Comments and Suggestions for Authors

Following the revisions made in accordance with the reviewers’ recommendations, the article has been thoroughly revised and improved in terms of structure, clarity, and scientific accuracy. As a result, it has been accepted for publication. These changes reflect the quality and relevance of the research, as well as its contribution to academic discourse.